# Low-Cost Chlorophyll Fluorescence Imaging for Stress Detection

**DOI:** 10.3390/s21062055

**Published:** 2021-03-15

**Authors:** Reeve Legendre, Nicholas T. Basinger, Marc W. van Iersel

**Affiliations:** 1Department of Horticulture, University of Georgia, Athens, GA 30602, USA; reeve.legendre@uga.edu; 2Department of Crop & Soil Sciences, University of Georgia, Athens, GA 30602, USA; nicholas.basinger@uga.edu

**Keywords:** chlorophyll fluorescence imaging, stress, herbicide, pixel intensity, PAM fluorometry, photosystem II

## Abstract

Plants naturally contain high levels of the stress-responsive fluorophore chlorophyll. Chlorophyll fluorescence imaging (CFI) is a powerful tool to measure photosynthetic efficiency in plants and provides the ability to detect damage from a range of biotic and abiotic stresses before visible symptoms occur. However, most CFI systems are complex, expensive systems that use pulse amplitude modulation (PAM) fluorometry. Here, we test a simple CFI system, that does not require PAM fluorometry, but instead simply images fluorescence emitted by plants. We used this technique to visualize stress induced by the photosystem II-inhibitory herbicide atrazine. After applying atrazine as a soil drench, CFI and color images were taken at 15-minute intervals, alongside measurements from a PAM fluorometer and a leaf reflectometer. Pixel intensity of the CFI images was negatively correlated with the quantum yield of photosystem II (ΦPSII) (*p* < 0.0001) and positively correlated with the measured reflectance in the spectral region of chlorophyll fluorescence emissions (*p* < 0.0001). A fluorescence-based stress index was developed using the reflectometer measurements based on wavelengths with the highest (741.2 nm) and lowest variability (548.9 nm) in response to atrazine damage. This index was correlated with ΦPSII (*p* < 0.0001). Low-cost CFI imaging can detect herbicide-induced stress (and likely other stressors) before there is visual damage.

## 1. Introduction

Plants naturally contain high levels of chlorophyll *a*, a fluorophore that is sensitive to a wide range of environmental stresses. Chlorophyll *a* fluorescence was first viewed in 1931 by Kautsky and Hirsch [1], nearly 90 years ago, by using a red-transmitting filter to view a plant that was moved from darkness to blue light. Since then, chlorophyll fluorescence has been developed into a tool that can quantify photosynthetic efficiency and detect a variety of stresses that affect the photosynthetic apparatus. Chlorophyll fluorescence is directly related to the photosynthetic efficiency of plants, because there are three possible fates of the energy of photons that have been absorbed by photosynthetic pigments. The energy form the photons can be used to drive the light reactions of photosynthesis (electron transport), be converted into heat, or be re-emitted as fluorescence by chlorophyll *a* [2]. The quantum yields of these three processes add up to one, so if the quantum yield of one process decreases, the quantum yield of one or both other processes will increase. The variability in chlorophyll fluorescence can thus be used as a measure of photosynthetic efficiency and primarily relates to changes in the efficiency with which photosystem II (PSII) uses the excitation energy from absorbed photons [3], because nearly all variable chlorophyll fluorescence comes from chlorophyll *a* surrounding PSII [4]. One of the primary strengths of chlorophyll fluorescence measurements is that many stresses can be detected before any signs of visible damage occur [5,6,7]. This is dependent on the type of stress incurred by the plant and has the potential to help minimize crop production problems through early detection of stressors that may negatively impact the crop, followed by appropriate recourse.

Chlorophyll fluorescence measurements can detect the physiological effects of a wide variety of stressors, including abiotic stressors such as high light, extreme temperatures, or drought [8,9,10,11,12,13], and chemical stressors like herbicides or heavy metals [11,14,15,16,17,18]. There are other applications in plant disease detection and screening for disease resistance in a variety of plant species [9,19,20,21,22]. The most common method to measure chlorophyll fluorescence uses pulse amplitude modulation (PAM) fluorometry. PAM fluorometers allow for measurement of chlorophyll fluorescence under ambient light conditions by pulsing low intensities of a measuring light. By measuring the amount of fluorescence emitted with and without the small pulses, PAM fluorometry can quantify the fluorescence induced by the measuring pulse only. PAM fluorometry requires measurements under both ambient light and during a saturating light pulse to determine the quantum yield of PSII (ΦPSII), which is a measure of the operating efficiency of PSII [23,24]. The saturating light pulse needs to be intense enough to fully saturate all photosynthetic reaction centers and typically has an intensity of 6000 to 10,000 µmol m^−2^ s^−1^, approximately 3 to 4 times to maximum photosynthetic photon flux density plants ever are exposed to under natural conditions. The requirement for such an extremely intense light pulse creates limitations, since it is infeasible to apply such a pulse to a large crop area, while also unsafe for people who might be exposed to such intense light. An additional measurement of chlorophyll fluorescence in the dark, using the measurement light only, is necessary to quantify changes in non-photochemical quenching (NPQ). The NPQ parameter describes how much heat dissipation is upregulated compared to a dark-adapted leaf, when heat dissipation is minimal. In addition to ΦPSII and NPQ, chlorophyll fluorometry can measure other, less common, photochemistry-related parameters: the quantum yield of non-photochemical energy dissipation in response to light exposure (ΦNPQ), and the quantum yield of other, non-light induced energy dissipation processes (ΦNO).

Many PAM fluorometers use fiber-optic cables to send the measuring light to a leaf and to capture the resulting fluorescence. Such systems are limited to point measurements, which cannot account for spatial variability that may be present across a leaf or canopy. Accurate assessment of the photosynthetic performance of a plant or crop can require many measurements to ensure that those samples are representative of the entire canopy.

A more recent development in monitoring chlorophyll fluorescence is chlorophyll fluorescence imaging (CFI), which also uses the principles of PAM fluorometry. Chlorophyll fluorescence imaging allows for visualization and quantification of the photosynthetic heterogeneity of an entire leaf or small canopy at once [25]. The PAM measuring light can be provided by pulsed lasers or light-emitting diodes (LEDs). When paired with the CCD camera and a control unit, the camera can be synced with the pulsed light to capture the fluorescence resulting from the measuring light. Since the chlorophyll *a* fluorescence spectrum peaks at 690 and 740 nm [26], CFI systems typically use a camera with a long-pass filter, while blocking shorter wavelengths. Once the images are captured, they are segmented and fluorescence parameters are calculated for each pixel, followed by visualization to produce chlorophyll fluorescence images [25]. Such CFI systems have been used successfully to detect damage from phytotoxic compounds [27], including herbicides [28]. However, due to the requirement for a saturating light pulse, commercially-available CFI systems are limited to individual leaves or small plants. In addition, commercially-available CFI systems require specialized electronics for image acquisition and processing and are expensive, while they often have low resolution.

Due to the complexity and expense of current CFI systems, a less expensive alternative that does not require PAM fluorometry for early stress detection would be beneficial, especially if it allows for measurements at larger scales. While the PAM-based CFI systems provide highly detailed, quantitative measurements, capturing images that simply display chlorophyll fluorescence may provide enough information to detect stress before it would otherwise be visible.

A simple, inexpensive CFI system can be assembled using a digital camera with a long pass filter, allowing red and/or far-red light to pass, a light secure enclosure, and blue light to drive photosynthesis [29]. There are limitations to such systems, as it is not possible to calculate the fluorescence parameters acquired using PAM fluorometers, like ΦPSII, NPQ, ΦNPQ or ΦNO [30]. However, a simple CFI system may be able to visualize plant stress. We conducted two studies to determine whether a simple imaging system is capable of visualizing changes in the two most common chlorophyll fluorescence parameters.

In the first study, we used a simple CFI system to capture chlorophyll fluorescence images of plants exposed to the herbicide atrazine to examine whether this system can be used for early stress detection, using pixel intensity as a direct measurement the fluorescence intensity. Atrazine (2-chloro-4-ethylamine-6-isopropyl-amino-*s*-triazine) is a triazine herbicide used primarily as a broad-spectrum control agent for broadleaf and grass weeds. Atrazine’s mode of action is as a PSII inhibitor that competitively binds to plastoquinone-binding proteins of PSII. Doing so prevents normal electron transport required for the light reactions of photosynthesis, which results in an inhibition of photosynthesis and extensive oxidative damage in chloroplasts [31,32,33,34]. The direct inhibition of PSII by atrazine makes it a good candidate for use in our study since inhibition of PSII activity, and thus photosynthetic electron transport, is expected to increase chlorophyll fluorescence. In addition to imaging the chlorophyll fluorescence, measurements of the leaf reflectance/fluorescence and ΦPSII were taken to look for correlations among these measurements. We hypothesized that the application of atrazine would result in an increased pixel intensity representative of damage to the photosynthetic apparatus of leaves and that this would be associated with a decrease in ΦPSII and an increase in measured leaf reflectance/fluorescence in the waveband where chlorophyll fluorescence emissions occur. In the second study, we determined whether our CFI system can visualize downregulation of NPQ after a plant that was exposed to relatively high light conditions is transferred to darkness. This typically results in downregulation of NPQ [35,36,37] and we therefore hypothesized that this would be accompanied by an increase in chlorophyll fluorescence.

## 2. Materials and Methods

### 2.1. Plant Materials and Herbicide Application

Three ‘Cora Punch’ vinca (*Catharanthus roseus*; Syngenta, Basel, Switzerland) plants in 10-cm pots filled with a peat-based substrate were purchased from a local garden center and used during this study. Two ‘Green Towers’ lettuce (*Lactuca sativa*) plants and two new guinea impatiens (*Impatiens hawkeri*) were also used, but results from those trials are not discussed in detail, since they responded essentially the same as the vinca plants. To induce stress, 250 mL of atrazine solution (AAtrex^®^ 4L, Syngenta; 1.33 mg of active ingredient/L) was applied as a soil drench resulting in an application rate equivalent to field rates (2.25 kg of active ingredient/ha). Atrazine is highly mobile in soil or peat-based substrates, allowing the root system of the plants to take up the atrazine, followed by distribution throughout the plant. 

For the duration of each run, one plant was placed inside a multispectral digital imaging system (Topview, Aris, Eindhoven, The Netherlands) and not moved after herbicide application. A cool-white LED panel (Cool white 225 LED ultrathin grow light panel, Yescom USA, City of Industry, CA, USA) was hung approximately 20 cm above the plant canopy and was used to drive photosynthesis and transpiration to promote herbicide uptake, movement, and physiological injury. The LED panel provided a photosynthetic photon flux density of approximately 175 µmol m^−2^ s^−1^ at the top of the canopy. Note that photosynthesis is a quantum-driven process and that light intensities therefore are typically reported as photon flux densities, rather than energy flux densities.

### 2.2. Reflectance/Fluorescence and ΦPSII

The fiber-optic cable of a spectrometer (Unispec Spectral Analysis System, PP Systems, Amesbury, MA, USA) was pointed at a leaf near the top of the canopy using a leaf clip to take leaf reflectance measurements. Reliable data could be detected at wavelengths from 450 to 770 nm due to the spectrum of the halogen bulb used by the spectrometer. Note that these leaf reflectance measurements in reality are a combination of the reflectance and chlorophyll fluorescence emitted from the leaves. The reflectance measurement immediately after herbicide application, before translocation of the atrazine to the leaves had occurred, was used as the baseline to normalize subsequent measurements. Since we were mainly interested in changes in chlorophyll fluorescence in response to the atrazine application, the normalized reflectance/fluorescence was averaged across the waveband of chlorophyll fluorescence (669.8–760.7 nm) to calculate the normalized average reflectance in the fluorescence spectrum (nARFS). The nARFS represents the change in the fluorescence spectrum from the initial timepoint.

A PAM fluorometer (Junior-PAM, Walz, Effeltrich, Germany) was used to take measurements of ΦPSII using a fiber optic cable aimed as closely as possible (<5 mm) at the location where the reflectance measurements were taken. Measurements using both the spectrometer and the PAM fluorometer were taken immediately prior to chlorophyll fluorescence imaging.

### 2.3. Chlorophyll Fluorescence Imaging

The cool-white LED panel was removed for ~1 min to facilitate digital imaging. For the chlorophyll fluorescence images, three plants (one plant per replication) were illuminated using blue LEDs, and a bandpass filter (650–740 nm) allowed only fluorescence to be captured by the monochrome digital camera in the Topview imaging system (Aris). Images were captured beginning immediately after herbicide application and subsequently every 15 min for 8 h. The monochrome images have an 8-bit resolution, resulting in a pixel intensity scale that spans from 0–255, where 0 represents a black pixel while 255 represents a white pixel, with varying shades of gray in between. Composite RGB images were taken at the same time using the multispectral imaging system, to compare to the fluorescence images and to see if symptoms of stress were visible in those RGB images (Appendix A). Each of the three replications were done on separate days.

### 2.4. Image and Data Analysis

To quantify the pixel intensity, chlorophyll fluorescence images were analyzed using the Fiji software package (www.fiji.sc, accessed on 14 March 2021). A 50 × 50 pixels square area, as close as possible to the area where the reflectance and ΦPSII measurements were taken, was selected and the average pixel intensity calculated. For statistical analysis, all data was analyzed using JMP Pro (version 15.0.0) using generalized linear models between each factor to check for correlations between nARFS, ΦPSII, and pixel intensity. Since our approach is more qualitative than quantitative, each replication was analyzed separately.

The standard deviation of the normalized average reflectance was calculated for all measured wavelengths to determine which wavelengths were best suited for the development of a fluorescence-based stress index (FBSI). Using a similar approach as already in use for other plant- or crop-based indices can facilitate the development of a sensor to detect stresses with a chlorophyll fluorescence-based stress signature.

### 2.5. Pixel Intensity and Heat Dissipation

A petunia (*Petunia* × *hybrida*) plant was moved from a greenhouse into the imaging system, where it was allowed to fully dark-adapt for 20 min, resulting in the opening of all photosystem II reaction centers and downregulation of heat dissipation. Dark-adapted ΦPSII was measured using the chlorophyll fluorometer. The plant was then exposed to white LED light with a photosynthetic photon flux density of approximately 550 µmol m^−2^ s^−1^ at the top of the canopy to induce upregulation of heat dissipation. After 15 min of light exposure, the LED fixture was removed from the imaging system and CFI images were collected at regular intervals over the following 15 min to determine whether the CFI images responded to the downregulation of heat dissipation. The plant was kept in the dark during this period, except for the brief light exposure required for the imaging. Following each image acquisition, the chlorophyll fluorometer was used to measure ΦPSII and NPQ, indicative of the upregulation of heat dissipation. NPQ was calculated based on the measured fluorescence during the saturating pulse after the initial dark acclimation (F_m_) and the measured fluorescence during the saturating pulses after the 15-minute exposure to light (F_m_’) as (F_m_ − F_m_’)/F_m_’.

The average pixel intensity and standard deviation of a 50 × 50 pixel area close to the fluorescence measurements was quantified using the Fiji software in all CFI images and we tested for correlations between pixel intensity, ΦPSII, and NPQ to determine whether CFI images could track changes in ΦPSII and NPQ.

## 3. Results

### 3.1. Chlorophyll Fluorescence Imaging

After application of the atrazine, areas with bright fluorescence became evident in the chlorophyll fluorescence images, beginning about two hours after the application. Increased fluorescence initially was evident along the midvein of the leaves and larger areas of the canopy fluoresced intensely as the atrazine spread through secondary veins (Figure 1A). 

Similar trends were seen in replications two and three (Appendix A) as well as lettuce and new guinea impatiens (results not shown). In the RGB images, no evidence of any damage was visible for the first eight hours after herbicide application (Figure 1B). Damage became visible only around 36 to 48 h after herbicide application (Appendix A).

### 3.2. Examination of Fluorescence from Reflectance Measurements

To confirm that the application of atrazine caused changes in chlorophyll fluorescence intensity, changes in the normalized reflectance/fluorescence were analyzed. These changes over time depended on wavelength, with wavelengths from 640 nm to 750 nm exhibiting a relatively large increase in reflectance/fluorescence over the duration of the experiment (Figure 2). The changes in the measured reflectance/fluorescence at different wavelengths over the course of the study were quantified by calculating the standard deviation of the normalized reflectance/fluorescence at each wavelength across all time periods, with a higher standard deviation indicating larger changes in the normalized reflectance/fluorescence. The largest variation in the normalized reflectance/fluorescence occurred at wavelengths above 650 nm, with peaks at 680 and 750 nm (Figure 3). The standard deviation of the normalized reflectance/fluorescence was low and similar at wavelengths from 450 to 600 nm.

### 3.3. Pixel Intensity, ΦPSII, and nARFS

Over the course of the experiment, the intensity of the 50 × 50 pixels area near the fluorometer and reflectance measurement site increased, indicative of increased chlorophyll fluorescence (Figure 4A). There was variability in the pixel intensity among the three replications, but the change in pixel intensity followed a similar pattern for all three plants. Pixel intensity was relatively stable for the first two to three hours, after which it gradually increased. Likewise, ΦPSII was relatively stable for the first two to four hours, after which the ΦPSII gradually decreased (Figure 4B). A slight dip in nARFS occurred during the initial 215 min after atrazine application in two of the three replications. This was followed by a strong increase in the nARFS during the remainder of the 8-h period (Figure 4C), during the same time that the pixel intensity increased and ΦPSII decreased.

### 3.4. Relationships between Average Pixel Intensity, ΦPSII, and nARFS

The primary goal of the study was to establish whether the pixel intensity acquired from the chlorophyll fluorescence images can be used to detect stress in the plant. Comparing the pixel intensity to ΦPSII verified that changes in pixel intensity are indeed indicative of physiological changes in the plants. There was a strong negative correlation between the pixel intensity and ΦPSII in all three replications (*r* < −0.90): as the pixel intensity decreased, the ΦPSII increased (Figure 5). This relationship was present, but quantitatively different among the three plants, indicative of the non-quantitative nature of our CFI approach.

To confirm that a change in pixel intensity was associated with a change in chlorophyll fluorescence, the relationship between the pixel intensity and nARFS was examined. A strong positive correlation between the pixel intensity and the nARFS was seen in all replications (*r* > 0.86; Figure 6), but once again this relationship differed among the three plants. There was a negative correlation between the nARFS and ΦPSII (*r* < −0.82, Figure 7), indicating that changes in nARFS were related to changes in the photosynthetic efficiency of the plant.

### 3.5. Development and Support for a Fluorescence-Based Stress Index

The FBSI (Equation (1)) was developed based on the standard deviation of the reflectance measurements at different wavelengths (Figure 3). The largest standard deviations in measured reflectance/fluorescence occurred at 698.3 nm and 741.2 nm, while the lowest standard deviation occurred at 548.9 nm. The reflectance measurements at 741.2 nm were chosen for use in the FBSI as they were better correlated with ΦPSII than the measurements at 698.2 nm, while the reflectance at 548.9 nm was used for normalization. The FBSI was defined using these wavelengths, where R is the reflectance at a specific wavelength:*FBSI = (R_741.2_ − R_548.9_)/(R_741.2_ + R_548.9_)*(1)

The FBSI was calculated for each timepoint and there were strong, negative correlations between the FBSI and ΦPSII for all three replications (*r* < −0.8, Figure 8), but once again these correlations were quantitatively different among the three plants.

### 3.6. Pixel Intensity, Heat Dissipation, and Quantum Yield Recovery in the Dark

The petunia plant fluoresced notably more brightly when dark-adapted than following a 15-minute exposure to a photosynthetic photon flux density of 550 µmol m^−2^ s^−1^ (Figure 9). The fluorescence intensity increased gradually following the light exposure, from a pixel intensity of 54.6 ± 3.8 at 30 s after the end of the light exposure to 59.6 ± 3.6 after 15 min. This increase in fluorescence was not clearly visible in the CFI images, but easily quantified. Pixel intensity was negatively correlated with NPQ and positively correlated with ΦPSII. Both correlations were highly significant, indicating that changes in pixel intensity in CFI images can be used to qualitatively follow downregulation of NPQ along with the associated increase in ΦPSII.

## 4. Discussion

Chlorophyll fluorescence has been used as a measure of stress for a number of different stressors including nutrient deficiencies [38,39,40], drought and water stress [41,42,43], extreme temperatures [44,45], harmful light conditions [46,47], herbicide-induced damage [14,15,16,17,48], and disease screening [9,19,20,21]. The transfer of electrons from photosystem II to the plastoquinone pool is the rate limiting step in the light reactions of photosynthesis. Until this electron transfer (photochemistry) has occurred and a new, oxidized plastoquinone has bound to the active site on PSII, additional electron transport through photosystem II is not possible without oxidative damage to the photosynthetic apparatus. Reaction systems incapable of normal electron transport are considered closed and closure of the reaction centers results in increased fluorescence from the chlorophyll *a* surrounding the reaction center. Since electron movement through photosystem I is faster than that through PSII, variable chlorophyll fluorescence comes from largely chlorophyll *a* surrounding PSII. Since a larger fraction of PSII reaction centers are closed under high light or certain stressful conditions, plants have evolved mechanisms to safely dissipate much of the excess light in the form of heat. This heat dissipation is triggered by acidification of the chloroplast lumen and the subsequent activation of pH-sensitive enzymes. This results in the conversion of the xanthophyll pigment violaxanthin into antheraxanthin and/or zeaxanthin. Antheraxanthin and zeaxanthin can trigger changes in the conformation of the light-harvesting complexes surrounding the reactions centers, resulting in the conversion of the absorbed light energy into heat, a process that also involves the PsbS protein in the PSII reaction center [49,50]. Since chlorophyll fluorescence, photochemistry, and heat dissipation are competitive processes, a change in photochemistry may not result in a change in chlorophyll fluorescence, if that change in photochemistry is accompanied by a corresponding change in heat dissipation.

Chlorophyll fluorometers use two different measurements to quantify ΦPSII, an initial measurement under ambient light (F_s_), followed by a second measurement during a saturating pulse of light (F_m_’). Both measurements are taken using the PAM approach, so that only the fluorescence induced by a weak modulating light is measured, rather than the total fluorescence emitted by the leaves. The ΦPSII is then calculated as (F_m_’ − F_s_)/F_m_’. Our approach differs fundamentally from the common approach, since we simple take images of the fluorescence emitted during exposure under low intensity blue LED light.

Traditional chlorophyll fluorometers can also detect changes in heat dissipation. To quantify changes in heat dissipation, a PAM-based measurement of a dark-adapted leaf during a saturating light pulse (F_m_) is required as well. Heat dissipation is down-regulated in dark-adapted leaves [49], thus increasing fluorescence. The magnitude of the decrease in fluorescence during a saturating light pulse in leaves exposed to actinic light as compared to the fluorescence from a dark-adapted leaf during a saturating light pulse is directly indicative of how much heat regulation has been upregulated in response to the actinic light.

Our CFI approach, where images are taken under low intensity blue light, cannot quantify ΦPSII, NPQ, or other commonly used fluorescence parameters. Although prior dark or light exposure is possible before taking the images, our CFI system cannot capture images of fluorescence using PAM or during a saturating light pulse. Doing so would greatly increase the complexity and cost of the imaging system. Therefore, our CFI approach differs fundamentally from traditional CFI imaging. However, as is clear from Figure 1, our rather simple approach can clearly visualize damage to the photosynthetic apparatus and can do so well before any visible symptoms occur. In cases where quantitative information is not required, our technique is easier to implement than traditional CFI.

In the current study, atrazine was used to induce stress by blocking electron transport through competitively binding to the plastoquinone binding site on PSII [51,52,53]. As most chlorophyll fluorescence is emitted from chlorophyll *a* surrounding PSII [4], it was expected that the atrazine application would cause a pronounced increase in fluorescence, which was indeed the case (Figure 1 and Figure 2). Using the CFI, the uptake of the herbicide could clearly be seen in the time lapse images (Figure 1A; Appendix A). The translocation of the atrazine throughout the plant was evident from an increasing number of bright pixels, initially along the leaf veins, and then throughout the entire leaves. CFI images showed damage long before any damage was visible. This agrees with previous studies that detected increases in chlorophyll fluorescence in response to a stress using PAM fluorometry, as well as more complex chlorophyll fluorescence imaging techniques before damage was visible [5,6,7,54]. The ability to detect stress early and take preventative measures to minimize damage may provide economic benefits to growers, as stress can decrease yield of many crops and lead to crop failure [55,56,57,58,59,60].

Chlorophyll *a* fluorescence has emission peaks at 680–690 nm and 730–740 nm [26,61]. This is consistent with the wavelengths where we observed the largest changes in normalized reflectance/fluorescence (Figure 2) and the largest standard deviation of those values over the course of the experiment (Figure 3). This confirms that the changes in the measured reflectance/fluorescence were indeed indicative of changes in chlorophyll *a* fluorescence and were likely not caused by changes in reflectance in response to the atrazine application.

The application of herbicide caused the pixel intensity to increase over time in all three replications (Figure 4A). Triazine herbicides like atrazine have been well documented to cause increases in fluorescence as powerful inhibitors of PSII [62,63,64,65,66]. ΦPSII gradually decreased after atrazine application (Figure 4B), which agrees with prior studies that have shown triazines to reduce ΦPSII [63,64,67]. The nARFS also increased after atrazine application (Figure 4C). The nARFS is an independent measurement to quantify the change in the fluorescence emitted by the leaf. Evidence of the nARFS being representative of actual chlorophyll fluorescence is provided by the relationship between the nARFS and ΦPSII. As ΦPSII decreased, the nARFS increased, indicative of increased chlorophyll *a* fluorescence (Figure 7). As previously established, the results agree with other studies that found chlorophyll *a* fluorescence is increased by the application of triazine herbicides like atrazine [62,63,64,65,66].

The objective of the first study was to determine whether the pixel intensity from a low-cost CFI system can be used to detect stress. The relationships between the pixel intensity, ΦPSII, and the nARFS in response to atrazine application all support that pixel intensity is indicative of chlorophyll fluorescence and damage induced by atrazine and thus can be used to detect stress. Pixel intensity from simple CFI images, to the best of our knowledge, has not previously been used to quantify chlorophyll fluorescence. Pixel intensity and the ΦPSII had a strong negative correlation (Figure 5), while a strong, positive correlation was found between pixel intensity and nARFS (Figure 6). Based on nARFS being representative of chlorophyll fluorescence (Figure 3 and Figure 4), this provides strong evidence that the pixel intensity is indicative of chlorophyll fluorescence.

While the correlations between pixel intensity, nARFS, and ΦPSII differed among runs, this is inconsequential for this qualitative approach; what matters is that there are highly significant correlations in all runs. That proves that changes in pixel intensity are indicative of changes in nARFS and ΦPSII, as is also evident from the time-lapse images (Figure 1; Appendix A). The differences in correlations among runs are likely due to differences among the leaves used, the exact position where data were collected (since there is great spatial variability), and the photosynthetic photon flux density at the measured leaf surface.

We developed a stress index, called the FBSI, based on two wavelengths: 741.2 nm and 548.9 nm, selected based on the standard deviation of the measure reflectance/fluorescence over time. Measuring this index requires leaf reflectance measurements, but it should be possible to develop a sensor that can specifically measure the FBSI, analogous to existing sensors that measure other dual-wavelength indexes, like the photochemical reflectance index or the normalized difference vegetation index [68,69]. For such a sensor, it is not critical that a narrow bandpass filter be used. As is evident from Figure 2, wavelengths from 480 to 600 nm are generally non-responsive to increased fluorescence, while wavelengths from 686 to 760 nm do respond strongly to increased fluorescence. Therefore, bandpass filters that cover these two wavebands could be used for a sensor to detect the FBSI remotely.

The objective of the 2nd study was to determine whether simple CFI imaging can be used to detect down-regulation of heat dissipation, as quantified by NPQ, after a plant has been exposed to light. Changes in pixel intensity, ΦPSII, and NPQ were followed during a 15-minute dark period and were strongly correlated. Given the competitive nature of photochemistry, heat dissipation, and chlorophyll fluorescence, this was expected. Reaction centers open rapidly in the absence of light, but enzymatic down-regulation of NPQ is much slower typically occurs over the course of minutes [49,50]. This is consistent with the gradual decrease in NPQ and increase in ΦPSII and pixel intensity that occurred during the 15 min following light exposure. As was the case with the atrazine-induced reduction in ΦPSII and associated increase in pixel intensity in the CFI images, changes in pixel intensity in the CFI images following light exposure of a plant provide a qualitative way to monitor down-regulation of NPQ and increase in ΦPSII in plants that are moved from light to darkness.

While the CFI system used in this study is not low cost, a system that performs an identical function can be constructed inexpensively. We expect that the hardware to construct an entire system can be reduced to ~100 USD [29]. All that is needed is a light-secure box (such as a grow tent), a digital camera with a long-pass filter (>690 nm), and blue LEDs to drive photosynthesis (450–495 nm). This setup works on the principle that in a light secure box, the only light available to illuminate the plant and drive photosynthesis is the blue light from the LEDs [29]. Because there is a filter in front of the camera that is blocking all light below 690 nm, the only light available for the camera to capture is chlorophyll fluorescence which has emission peaks at 680 nm and 750 nm. While some of the chlorophyll fluorescence will be excluded from the images, enough fluorescence is available for imaging. Such systems also have applications outside of stress detection by making it easier to distinguish the plants from the background during image analysis, which has a myriad of applications in other research involving plant growth and development [70]. While it is apparent that the low-cost system can easily detect stress induced by atrazine, further research is necessary to determine whether this system is able to detect other biotic and abiotic stressors. Preliminary findings suggest that the technique can detect cold damage, but not heat damage (unpublished data).

An ongoing study shows that damage caused by some, but not all, herbicides, is easier to detect using CFI than RGB imaging and that herbicide-resistance is easily detected. Glufosinate, for example, causes damage that is clearly visible in CFI images, while symptoms in RGB images are subtle. The glufosinate-resistant cultivar “SH7418LL” shows no damage in either CFI or RGB images, while the sensitive cultivar “Benning shows clear damage in CFI images at 24-h after application (Appendix A). Glufosinate inhibits the enzyme glutamine synthetase, which is present in chloroplasts. Inhibition of glutamine synthetase interferes with normal amino acid metabolism and can lead to the buildup of ammonium in chloroplasts, a toxic compound that interferes with the production of adenosine triphosphate, an energy-rich molecule required for photosynthesis [71].

It is not clear at this time which biotic and abiotic stresses can be detected using this technique, but it seems likely that direct or indirect effects on the light reactions of photosynthesis are a requirement for detection by CFI.

## Figures and Tables

**Figure 1 sensors-21-02055-f001:**
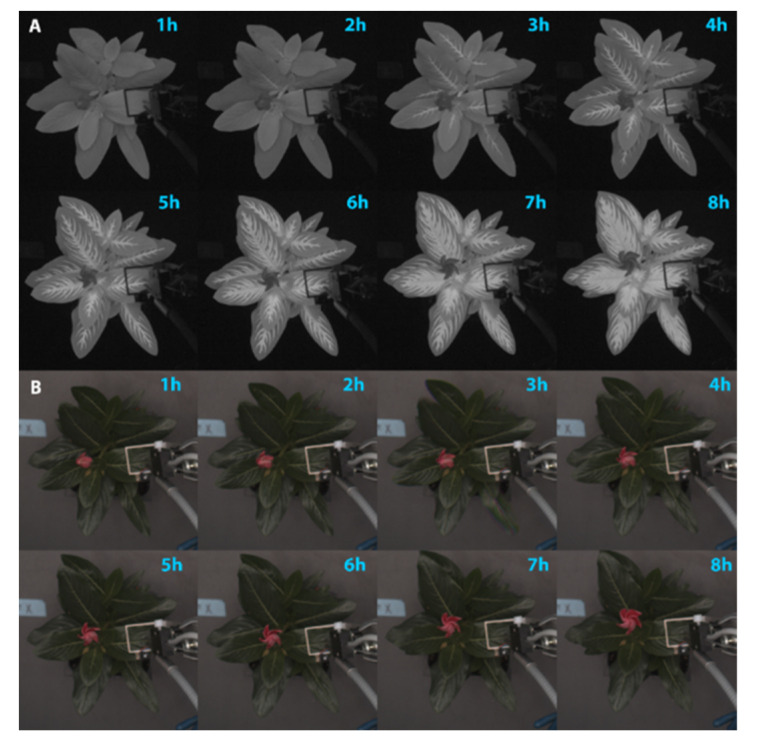
The effect of atrazine drench application on: (**A**) chlorophyll fluorescence and (**B**) visual appearance of *Catharanthus roseus*. Time since herbicide application is indicated in the upper righthand corner of each image.

**Figure 2 sensors-21-02055-f002:**
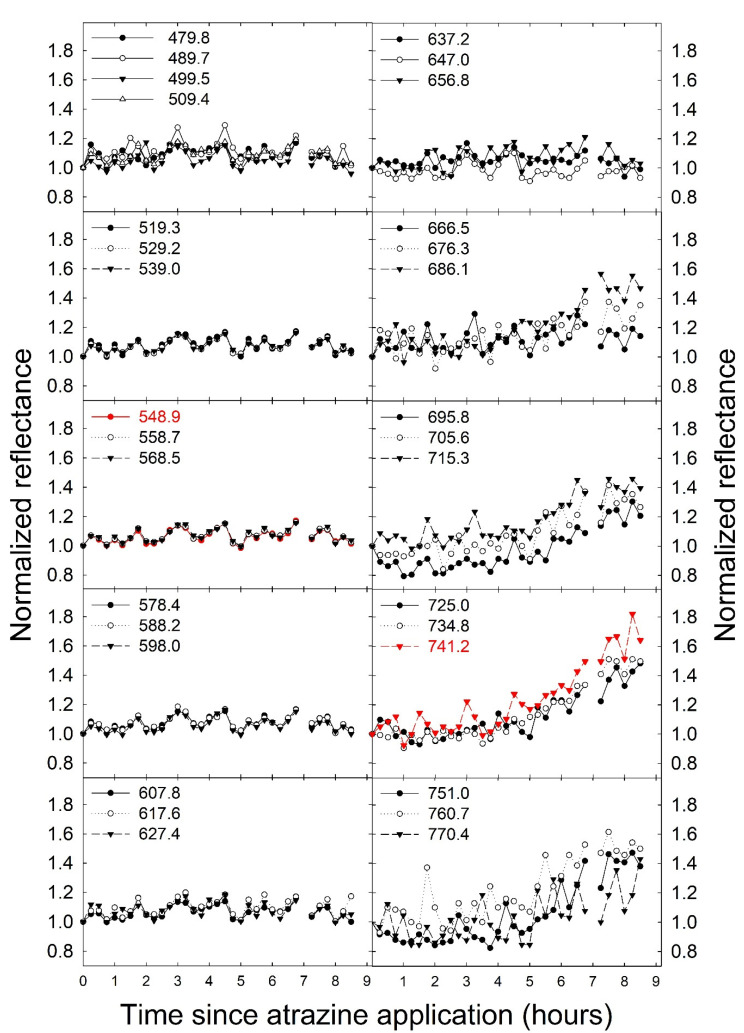
The normalized reflectance/fluorescence (normalized to the measured value at time 0) of *Catharanthus roseus* at different wavelengths as a function of the time since atrazine application. Values in the figure legends indicate the wavelength at which the reflectance/fluorescence was measured (nm). The two data sets in red indicate the wavelengths used for the development of the fluorescence-based stress index.

**Figure 3 sensors-21-02055-f003:**
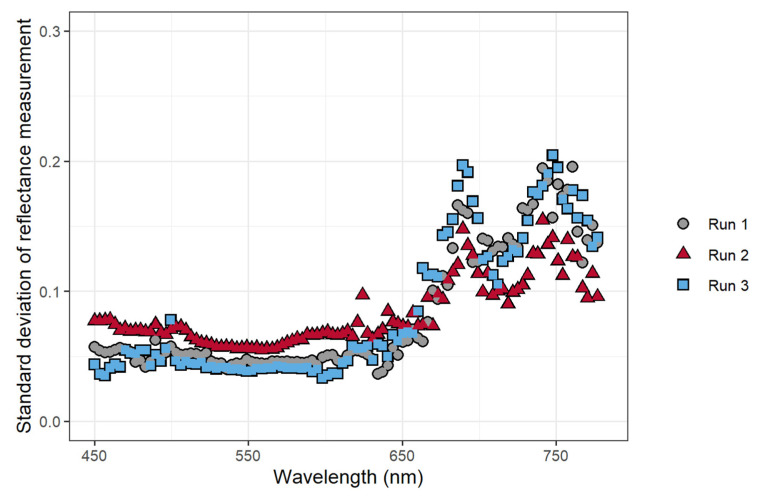
The effect of atrazine application on the standard deviation of the normalized average reflectance/fluorescence of *Catharanthus roseus* as a function of the wavelength. Data used to calculate the standard deviation was taken every 15 min after herbicide application for 8 h (*n* = 32).

**Figure 4 sensors-21-02055-f004:**
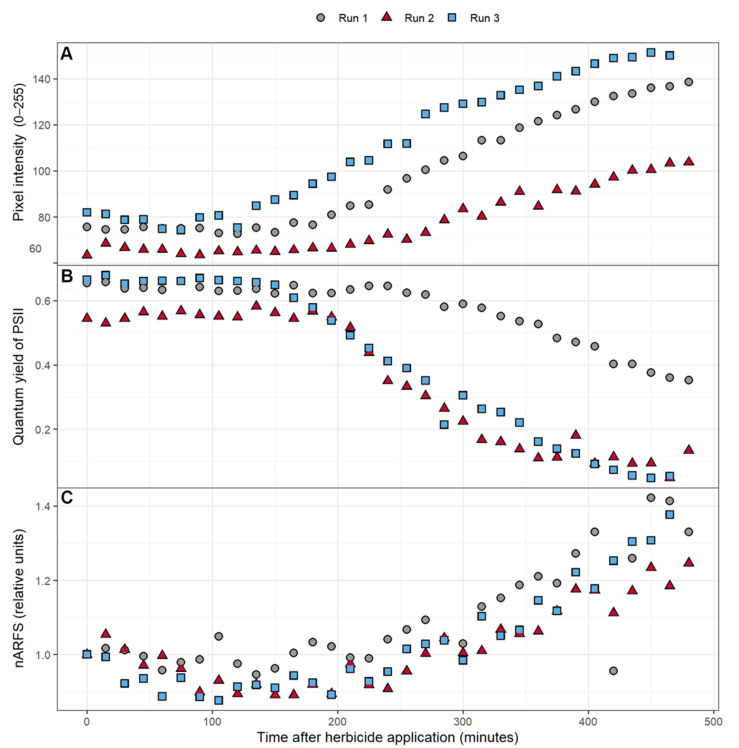
The effect of atrazine application on: (**A**) the average pixel intensity of a 50 pixel by 50 pixel area near where the other measurements were taken, (**B**) the quantum yield of PSII, (**C**) and the normalized average reflectance of the reflectance spectrum (nARFS) of *Catharanthus roseus* over an 8-h period. The nARFS was determined by averaging the normalized reflectance in the chlorophyll fluorescence spectrum (669.8–760.7 nm) and dividing that value by the value from the first reflectance measurement after herbicide application.

**Figure 5 sensors-21-02055-f005:**
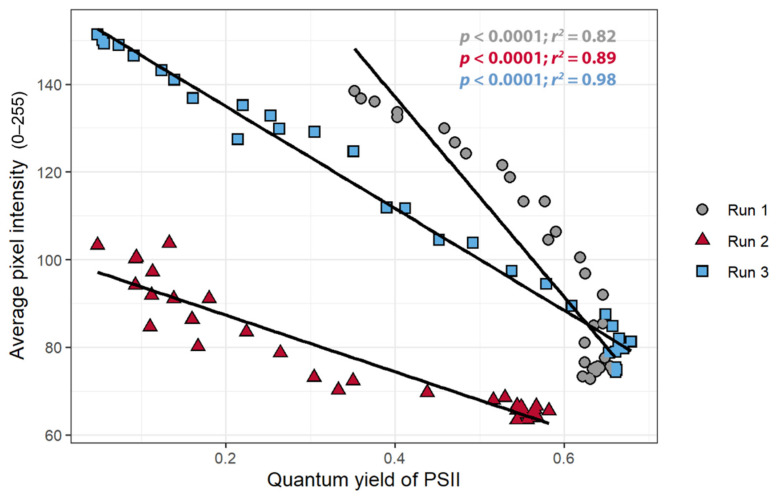
The relationship between the quantum yield of photosystem II and the average pixel intensity in chlorophyll fluorescence images of *Catharanthus roseus* treated with atrazine.

**Figure 6 sensors-21-02055-f006:**
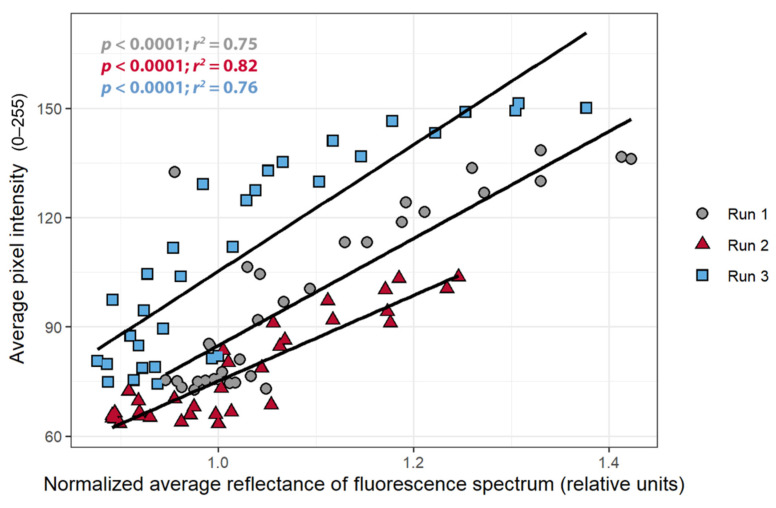
The relationship between normalized average reflectance of fluorescence spectrum and the average pixel intensity of *Catharanthus roseus* treated with atrazine. The average pixel intensity was taken from the chlorophyll fluorescence images in a 50 × 50 area near the spot of the reflectance measurements.

**Figure 7 sensors-21-02055-f007:**
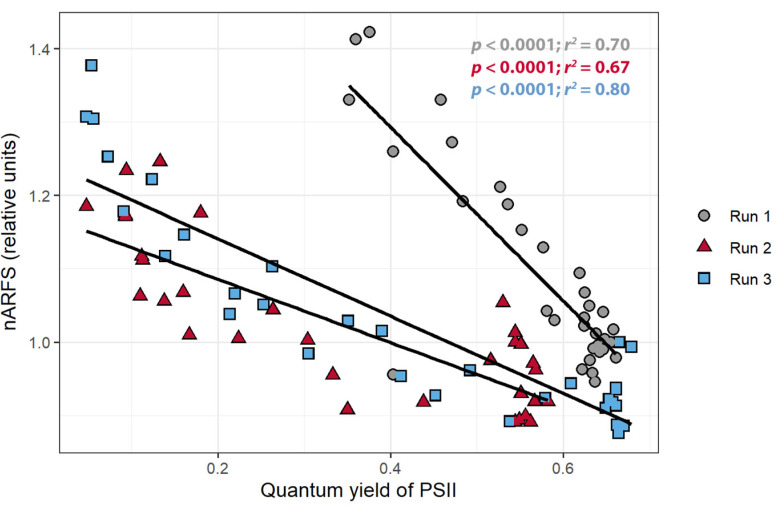
The relationship between the normalized average reflectance of fluorescence spectrum (nARFS) and the quantum yield of photosystem II of *Catharanthus roseus* treated with atrazine.

**Figure 8 sensors-21-02055-f008:**
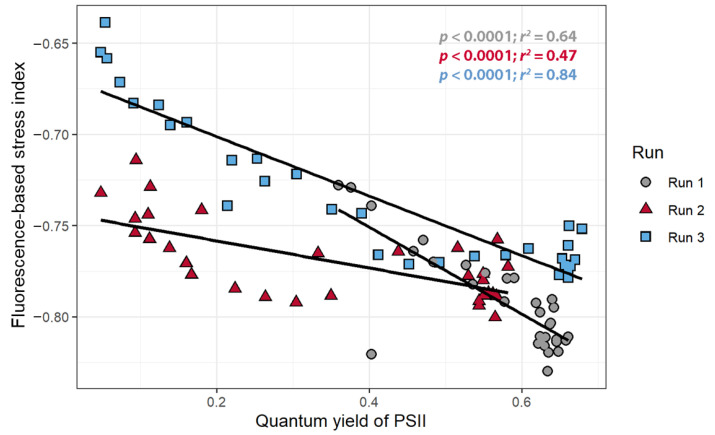
The relationship between the fluorescence-based stress index and the quantum yield of photosystem II of *Catharanthus roseus* treated with atrazine.

**Figure 9 sensors-21-02055-f009:**
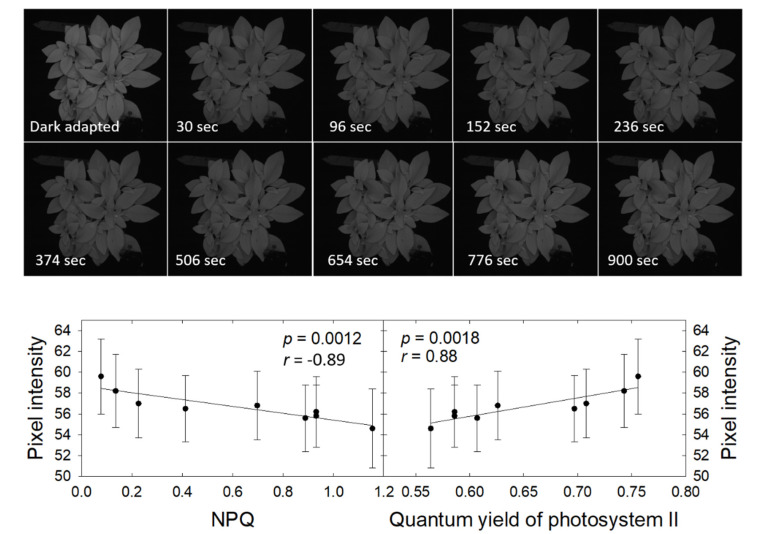
Chlorophyll fluorescence images of *Petunia × hybrida* following dark-adaptation and at different times following 15 min at a photosynthetic photon flux density of 550 µmol m^−2^ s^−1^ and subsequent return to darkness (**top**). The relationship between pixel intensity and measured NPQ (**bottom left**) and quantum yield of photosystem II (**bottom right**) (measured in the white square). The pixel intensity of the dark-adapted plant was 80.5 ± 5.0.

## Data Availability

Data and images used in this study are available at https://drive.google.com/drive/u/1/folders/1jSLTtcmE-3ogMUMd9QUp4VyVE2x46zR1, accessed on 14 March 2021.

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
