# Peer review of "Low-Cost Chlorophyll Fluorescence Imaging for Stress Detection"

_sensors, 2021, doi:10.3390/s21062055_

Round 1

Reviewer 1 Report

In this manuscript, the authors proposed a simple Chlorophyll fluorescence imaging (CFI) device for plants, which can use the pixel intensity in the fluorescence image to qualitatively determine whether the plants are damaged by atrazine. The device has practical applications in the field of plants. This manuscript is well organized and presented. My suggestion is that the manuscript can be accepted after a revision. My specific comments are listed as following:

  1. Only the atrazine is used in the manuscript and other types of herbicides are not mentioned. Does the device only work for some certain types of herbicide?
  2. The manuscript said, “Note that these leaf reflectance measurements in reality are a combination of the reflectance and chlorophyll fluorescence emitted from the leaves. The reflectance measurement immediately after herbicide application, before translocation of the atrazine to the leaves had occurred, was used as the baseline to normalize subsequent measurements.” After the injection of herbicide, the leaf morphology and reflectivity may certainly change. This factor should be considered.
  3. It would be better to have a schematic diagram of the CFI system used in the paper.
  4. The experimental data of the 7th hour in Figure 2 is missing. Please check it.
  5. There are 63 references in total, but only 8 in the last five years. The proportion of literatures in recent years is not enough.
  6. There are two ‘3.3’ headings in the article. No doubt, the format of the article needs to be checked as well.

Author Response

  • Two paragraphs and a supplementary figure were added at the end of the discussion to address the applicability of the CFI system on other herbicides. More importantly, we emphasized more strongly that applications are not limited to herbicides, but likely apply to all processes that affect photosynthetic electron transport.
  • In order to clarify that the changes in the reflectance measurements were due to chlorophyll fluorescence and likely not caused by changes in reflectance, clarification was added to the end of the first paragraph on page 12. The spectral changes in reflectance/fluorescence that were seen are entirely consistent with the spectrum of chlorophyll fluorescence.

  • WEW have added a diagram of the imaging system as a supplementary figure and explained how that system was used to obtain the images we used.
  • The 7th hour was omitted because of technical problems which resulted in clearly incorrect data.

  • More recent references were added throughout the text.

  • The subheading numbering has been corrected.

Reviewer 2 Report

The message the authors want to convey is quite clear, science is sound, I recommend acceptance of the manuscript as is. 

Author Response

Reviewer had no suggested changes

Reviewer 3 Report

Dear Authors,

Congratulations on your work! Your paper has a significant scientific value and purpose for the detection Chlorophyll Fluorescence thoufh a low cost method. Also, the paper is very well written and, according to my opinion, no correction is needed regarding the English language. I suggest, however, for you to explain a little more in detail the statistical method you used for the analysis of the data, the number of plants tested and replications per experiment. I believe it would increase the strength of the data you are presenting.

All the Best.

Author Response

To better explain the statistical methods, section 2.3 has been broken into 2.3 and 2.4 with a more detailed description of the statistical analysis in section 2.4.